# A Non-Cell-Autonomous Mode of DNA Damage Response in Soma of *Caenorhabditis elegans*

**DOI:** 10.3390/ijms23147544

**Published:** 2022-07-07

**Authors:** Zhangyu Dai, Wenjing Zhang, Mengke Shang, Huangqi Tang, Lijun Wu, Yuejin Wu, Ting Wang, Po Bian

**Affiliations:** 1Key Laboratory of High Magnetic Field and Ion Beam Physical Biology, Hefei Institutes of Physical Science, Chinese Academy of Sciences, Hefei 230031, China; dzy1124@mail.ustc.edu.cn (Z.D.); 15655300772@163.com (W.Z.); candiseven@126.com (H.T.); yjwu@ipp.ac.cn (Y.W.); 2Science Island Branch of Graduate School, University of Science and Technology of China, Hefei 230026, China; 3Institute of Physical Science and Information Technology, Anhui University, Hefei 230601, China; smk0717@163.com (M.S.); ljw@ipp.ac.cn (L.W.); 4Teaching and Research Section of Nuclear Medicine, School of Basic Medical Sciences, Anhui Medical University, Hefei 230032, China

**Keywords:** DNA damage response, non-cell-autonomy, ionizing radiation, CPR-4, *C. elegans*

## Abstract

Life has evolved a mechanism called DNA damage response (DDR) to sense, signal and remove/repair DNA damage, and its deficiency and dysfunction usually lead to genomic instability and development of cancer. The signaling mode of the DDR has been believed to be of cell-autonomy. However, the paradigm is being shifted with in-depth research into model organism *Caenorhabditis elegans*. Here, we mainly investigate the effect of DDR activation on the radiosensitivity of vulva of *C. elegans*, and first found that the vulval radiosensitivity is mainly regulated by somatic DDR, rather than the DDR of germline. Subsequently, the worm lines with pharynx-specific rescue of DDR were constructed, and it is shown that the 9-1-1-ATR and MRN-ATM cascades in pharynx restore approximately 90% and 70% of vulval radiosensitivity, respectively, through distantly regulating the NHEJ repair of vulval cells. The results suggest that the signaling cascade of DDR might also operate in a non-cell autonomous mode. To further explore the underlying regulatory mechanisms, the *cpr-4* mutated gene is introduced into the DDR-rescued worms, and CPR-4, a cysteine protease cathepsin B, is confirmed to mediate the inter-tissue and inter-individual regulation of DDR as a signaling molecule downstream of 9-1-1-ATR. Our findings throw some light on the regulation of DNA repair in soma of *C. elegans*, and might also provide new cues for cancer prevention and treatment.

## 1. Introduction

Life is continuously subjected to exogenous and endogenous DNA damage. To counter this threat, a mechanism called DNA damage response (DDR) has been evolved [1]. The DDR comprises a set of signaling pathways for sensing and signaling the DNA damage. Depending on their distinct positions and functions, proteins involved in the DDR pathway are classified as sensors that detect the damage, transducers that propagate the damage signals, and effectors that elicit some specific biological responses to preserve genetic integrity [1]. DNA damage is initially sensed by a variety of damage-specific DNA-binding proteins, such as RAD9-HUS1-RAD1 (9-1-1) and MRE11-RAD50-NBS1 (MRN) complexes. The damage-sensing signals are further propagated through two key transducers ataxia-telangiectasia mutated (ATM) and ATM-Rad3-related (ATR) [2]. Downstream of these transducers are effector molecules such as p53 and CDC25, which elicit cycle checkpoints, DNA repair, and cell apoptosis to cope with DNA damage. The DDR operates in diverse biological settings and its deficiency and dysfunction usually lead to genomic instability that is a major factor driving the onset and progression of carcinogenesis [3,4]. On the other hand, DDR deficiency has recently been considered to be an important determinant of tumor immunogenicity. There is growing evidence that DDR-targeted therapies can increase the antitumor immune response by promoting antigenicity, adjuvanticity, and reactogenicity of tumor cells [5].

Current knowledge of DDR functions is mainly based its cell autonomous mode [6]. It has been reported that in the model organism *Caenorhabditis elegans*, extracellular signals regulate some key components of DDR in germline [7,8,9,10,11]. However, it is unclear whether DDR own signaling process is of non-cell autonomy, i.e., DDR upstream signaling cascades in a cell can substantially regulate its downstream responses in another cell. This is relevant to the tumor micro-environment where normal cells interact with DDR-deficient tumor cells. Notably, in radiation-induced bystander effects (RIBE), irradiated cells can produce some DDR-dependent bystander signals, and then cause cycle arrest, DNA repair, or apoptosis in bystander cells. However, these downstream responses in bystander cells are typically thought to be autonomously induced by their own DDR, which is initiated de novo by the bystander signal-induced DNA damage [12,13]. Research on DDR in *C. elegans* is mainly carried out with the tissue model of germline [14], but there are also unique responses to DNA damage in somatic tissues of worms, in which activation of the apical components of DDR is suppressed at the transcriptional level, whereas their ability to repair DNA damage is well maintained. This has led to a long-standing confusion about how DNA repair apparatuses are upstream controlled in soma of *C. elegans* [15]. On the other hand, these somatic features of worms also allow us to explore some non-canonical schemes of DDR.

In classical radiotherapy, reproductive cell death, or known as clonogenic death, is the primary mechanism that achieves tumor control [16,17]. The vulva of *C. elegans* is a well-established tissue model for radiation-induced reproductive cell death [18]. In this model, three vulval precursor cells (VPCs), P5.p, P6.p, and P7.p, develop to 22 cells after three rounds of synchronized divisions, eventually maturing a functional vulva. Ionizing radiation can induce reproductive death among the dividing progeny cells of VPCs, and results in the malformation of vulva, primarily a protruding phenotype when exposed to lower radiation doses. As a typical somatic tissue of *C. elegans*, the vulva shows an undetectable apical activity of DDR [15], but its DNA repair ability is well retained [19,20]. Due to the absence of apical activity of DDR in vulva, the vulval radiosensitivity should not be changed when the apical components of DDR of *C. elegans* is genetically deficient. However, it is surprising that the vulva exhibits enhanced radiosensitivity in worms impaired in some apical component-encoding genes such as *hus-1*, *atm-1*, *mrt-2*, and *clk-2* [18,19,21]. This implies that the upstream portion of DDR might regulate the DNA repair of the vulva in an inter-tissue manner. The aim of this study is to experimentally verify the non-cell autonomous mode of DDR.

## 2. Results

### 2.1. Participation of Somatic DDR in Regulating Vulva Radiosensitivity

In order to determine whether the DDR of germline participates in regulating vulval radiosensitivity, we adopted *glp-1* worms, which gonads contain no germ cells when cultured at 25 °C [22], as shown in Figure 1A. Mutation of the *glp-1* gene *per se* did not cause vulval malformation (Table A3). After 100 Gy of γ-irradiation, no significant difference in vulval malformation was observed between N2 and *glp-1* worms (*p* > 0.05) (Figure 1B), indicating that vulval radiosensitivity might not be regulated by DDR of the germline. Therefore, we further examined whether DDR in somatic cells of *C. elegans* is involved in this regulation. MRT-2, RAD50, and MRE-11 are members of the 9-1-1 and MRN complexes apical in the DDR pathway [3]. In the present study, we specifically knocked down their expressions in somatic cells using *ppw-1* worms [23]. It was shown that RNAi of the *mrt-2*, *rad-50*, and *mre-11* genes enhanced vulval radiosensitivity (in all cases, *p* < 0.05) (Figure 1C), preliminarily suggesting a functional regulation of somatic DDR on the vulval radiosensitivity. Considering the sin somatic cells of *C. elegans*, we put forward a hypothesis that a portion of somatic cells (possibly all) might jointly contribute to a functional DDR that can regulate their DNA repair in a non-cell-autonomous manner.

### 2.2. Inter-Tissue Regulation of Pharyngeal ATR Cascade on Vulva Radiosensitivity

In order to test the above hypothesis, we constructed integrant worm lines transgenic for the P*myo-2* driven *hus-1* gene and crossed them into the *hus-1* mutation background (Figure 2A). In the resulting line BPL1121: P*myo-2::gfp::hus-1; hus-1(op241)*, the function of the DDR in the pharynx is specifically rescued from the *hus-1* mutation background. The BPL1121 worms exhibited a reduced vulval radiosensitivity compared with *hus-1* worms after 50 Gy, 100 Gy, and 150 Gy of γ-irradiation (*p* < 0.01 for 50 and 100 Gy, and *p* > 0.05 for 150 Gy), and the most significant was at 100 Gy, as shown in Figure 2B. Thus, the dose of 100 Gy was applied in the following experiments, unless otherwise specified. We also excluded the possible influence of the transgene-integrated locus on vulval radiosensitivity by examining two other independent worm lines, BPL1122: P*myo-2::gfp::hus-1; hus-1(op241)* and BPL1123: P*myo-2::gfp::hus-1; hus-1(op241)* (Figure A1A). To exclude the additional influence of P*myo-2::hus-1* over-expression itself, we detected vulval radiosensitivity of worm line BPL1111: P*myo-2::gfp::hus-1; N2*, in which P*myo-2::hus-1* is expressed under a wild type background. The vulval radiosensitivity of BPL1111 worms was not significantly different from that of N2 worms (Figure A1B). This was also true for the negative control worm lines BPL1100: P*myo-2::gfp; N2* and BPL1101: P*myo-2::gfp; hus-1(op241)* (Figure A1C,D), which are identical to BPL1111 and BPL1121 worms, except for absence of the transgenic *hus-1* gene. We also knocked down the expressions of the *mrt-2* and *atl-1* genes in BPL1121 worms, which encode another member of the 9-1-1 complex and an ortholog of mammalian ATR, respectively. It was found that RNAi of the *mrt-2* and *atl-1* genes significantly enhanced the vulval radiosensitivity of BPL1121 worms (in both cases, *p* < 0.05) (Figure 2C), indicating that HUS-1 in the pharynx might also function through the classic 9-1-1-ATR signaling cascade. These results suggest that pharyngeal 9-1-1-ATR cascade might regulate vulval radiosensitivity in an inter-tissue manner.

### 2.3. Inter-Individual Regulation of Pharyngeal ATR Cascade on Vulval Radio Sensitivity

At this point, it is not completely excluded that the reduced vulval radiosensitivity in BPL1121 worms is due to leaky expression of P*myo-2::hus-1* in vulval cells, although GFP fluorescence was not observed in vulval cells of BPL1121 worms (Figure 2A). In order to test this possibility, we co-cultured BPL1121 and *hus-1* worms at a ratio of 1:1 (Figure 3A), and found that the irradiated BPL1121 worms released a damage signal into medium, and reduced vulval radiosensitivity of the co-cultured *hus-1* worms (*p* < 0.01), as shown in Figure 3B. The worms with null mutation in the *hus-1 (op244)* gene were also adopted for the co-culture experiment, and their vulval radiosensitivity was likewise reduced (*p* < 0.01) (Figure 3B). The *hus-1* worms co-cultured with sham-irradiated BPL1121 and irradiated BPL1101 showed no change in the vulval radiosensitivity compared with the *hus-1* worms cultured alone (Table A2, and Figure A2). These results exclude the possible influence of leaky expression of P*myo-2::hus-1*, if any, on the vulval radiosensitivity. However, the co-culture results also cause a new confusion that whether in internal or external manners, the pharynx-derived damage signal reaches to the vulva in BPL1121 worms, as shown schematically in Figure 3C. To address this question, we changed the ratio of BPL1121 and *hus-1* worms in the co-culture experiments. When the ratio of BPL1121 and *hus-1* worms was changed from the original 1:1 to 1:5 and 1:10, respectively, the vulval radiosensitivity of the BPL1121 worms was still similar to that of BPL1121 worms co-cultured at a ratio of 1:1 (in both cases, *p* > 0.05), whereas the vulval radiosensitivity of co-cultured *hus-1* worms was not significantly reduced compared with that of *hus-1* worms at a ratio of 1:1 (in both cases, *p* < 0.01), as shown in Figure 3D. These results suggest that in BPL1121 worms the damage signal might be internally transmitted/cascaded from pharynx to vulva, at least under the condition of low-density culture.

### 2.4. Regulation of Pharyngeal 9-1-1-ATR Cascade on Non-Homologous End Joining of Vulva

Except for the apoptosis [18], the cycle checkpoint and DNA repair both can modulate the radiosensitivity of vulva [1]. In order to determine which is the downstream target of pharyngeal 9-1-1-ATR, we first measured vulval radiosensitivity of worms with mutations in the *cdc-25.1*, *cdc-25.3* and *wee-1.1* genes, which encode three key components of the G1/S and G2 checkpoints, respectively [24]. As shown in Figure 4A, these mutants did not exhibit higher vulval radiosensitivity compared with N2 worms (in all cases, *p* > 0.05), suggesting that cycle checkpoints might not participate in the responses of vulva to DNA damage. Next, we examined the role of DNA repair in the vulva by employing the worms impaired in DNA repair pathways. It has been reported that for γ-irradiation non-homologous end joining (NHEJ) and mismatch repair (MMR) are the principal executors of DNA damage repair in vulval cells [18,19]. Therefore, we co-cultured BPL1121 worms with *cku-80* and *lig-4* worms (NHEJ), and *msh-2* worms (MMR). It was found that the co-cultured *lig-4* and *cku-80* worms exhibited higher vulval radiosensitivity relative to the co-cultured N2 worms (in both cases, *p* < 0.01), although it was lower relative to their controls cultured alone (in both cases, *p* < 0.05). However, the vulval radiosensitivity of *msh-2* worms was reduced to that of N2 worms (*p* > 0.05), as shown in Figure 4B. These results suggest that pharyngeal 9-1-1-ATR cascade might mainly regulate the NHEJ of vulva.

### 2.5. The Role of CPR-4 in Mediating Non-Cell-Autonomy of Pharyngeal ATR Cascade

In order to identify the chemical nature of the signaling molecule(s) that mediates the non-cell-autonomy of DDR, the conditioned medium of irradiated BPL1121 worms was treated with DNase, RNase, and trypsin, respectively. As shown in Figure 5A, the vulval radiosensitivity of *hus-1* worms was not significantly reduced when cultured in conditioned medium treated with trypsin (*p* > 0.05), indicating that the signaling molecule(s) is a type of protein. It has been reported that CPR-4, a cysteine protease cathepsin B, mediates radiation-induced bystander effects in *C. elegans* [9]. In order to further evaluate its role in the non-cell autonomy of DDR, we crossed BPL1121 worms into a *cpr-4* mutation background, generating a worm line BPL1131: P*myo-2::gfp::hus-1; hus-1(op241); cpr-4(ok3413)*. The BPL1131 worms exhibited higher vulval radiosensitivity relative to BPL1121 worms (*p* < 0.01) (Figure 5B). Moreover, when *hus-1(op241)* worms were co-cultured with BPL1131, their vulval radiosensitivity was similar to that of worms cultured alone (*p* > 0.05) (Figure 5C). Consistently, the vulval radiosensitivity of *hus-1* worms was not reduced when cultured in BPL1121 conditioned medium treated with CPR-4 inhibitors baicalein and CA-074 [25] (in both cases, *p* > 0.05) (Figure 5D). These results suggest that CPR-4 might act as a distant signaling molecule to mediate the inter-tissue and inter-individual regulations of pharyngeal 9-1-1-ATR cascade.

### 2.6. Systemic Regulation of Pharyngeal 9-1-1-ATR Cascade

The above results also raised a question regarding whether inter-tissue mode of the pharyngeal 9-1-1-ATR cascade is specific only to the vulva. DNA damage can usually lead to growth retardation in the forms of cell death and repressed cell reproduction [26]. In order to address this question, we first measured the body growth of BPL1121 worms after γ-irradiation. Compared with *hus-1* worms, the repression of body growth of BPL1121 worms was only slightly attenuated (*p* < 0.01). However, the percentage of BPL1121 worms with body length of less than 1mm was significantly reduced compared with the *hus-1* worms (Figure 6A). Therefore, we further examined the effect of pharyngeal 9-1-1-ATR cascade on whole-worm development with the formation of U-shaped gonads as an indicator of L3/L4 transition [27], as shown in Figure 6B. After γ-irradiation, the developmental repression of BPL1121 worms was significantly alleviated compared with *hus-1* worms (*p* < 0.05), as shown in Figure 6C. These results suggest that regulation of the pharyngeal 9-1-1-ATR cascade might also be systemic.

### 2.7. Inter-Tissue Regulation of the Pharyngeal ATM Cascade

Another open question is whether the MRN-ATM cascade is also of non-cell-autonomy in somatic cells of *C. elegans*. To address this, we constructed an integrant line BPL1221: P*myo-2::gfp::atm-1; atm-1(gk186)* (Figure 7A). Compared with *atm-1* worms, the BPL1221 worms exhibited a reduced vulval radiosensitivity (*p* < 0.01), as shown in Figure 7B. Another independent line BPL1222: P*myo-2::gfp::atm-1; atm-1(gk186)* showed the similar change (Figure A3A). For the control lines BPL1211: P*myo-2::gfp::atm-1; N2* and BPL1201: P*myo-2::gfp; atm-1(gk186)*, their vulval radiosensitivity was similar to N2 and *atm-1* worms, respectively (Figure A3B,C). In the co-culture experiments for BPL1221 and *atm-1* worms, the vulval radiosensitivity of *atm-1* worms was also significantly reduced (*p* < 0.01) (Figure 7C). The RNAi of the *mre-11* gene, encoding a member of the MRN complex upstream of ATM-1, enhanced the vulval radiosensitivity of BPL1221 worms (*p* < 0.01) (Figure 7D). These results suggest that pharyngeal MRN-ATM cascade might also operate in non-cell autonomous manner.

Next, we crossed the BPL1221 into the *cpr-4* mutation background, and the resulting line BPL1231: P*myo-2::gfp::atm-1; atm-1(gk186); cpr-4(ok3413)* exhibited a vulval radiosensitivity similar to that of BPL1221 worms (*p* > 0.05) (Figure 7E). Moreover, the vulval radiosensitivity of *atm-1* worms was still reduced when co-cultured with BPL1231 worms (*p* < 0.01) (Figure 7F). The vulval radiosensitivity of *atm-1* worms was similarly reduced when cultured in the conditioned medium treated with CPR-4 inhibitors baicalein or CA-074 (in both cases, *p* < 0.01) (Figure 7G). These results suggest that the pharyngeal MRN-ATM cascade might not employ CPR-4 as a downstream signaling molecule.

## 3. Discussion

In the present study, we discovered a distinct mode of non-cell-autonomy of DDR by demonstrating inter-tissue and inter-individual regulations of pharyngeal DDR on DNA repair in the vulva of *C. elegans*. It is well known that the DDR in germline of *C. elegans* is intact and active, while the upstream activity of DDR is absent in somatic cells, including the vulval cells [14]. If the radiosensitivity of vulva is regulated by DDR in otherwise tissues, the DDR of germline is undoubtedly the first suspicion. However, our results did not support this assumption (Figure 1), implying that the vulval radiosensitivity might be regulated by the somatic DDR in a non-cell autonomous manner. In order to confirm it, we put forward a experimental strategy that a functional DDR is retained in a given set of somatic cells, and its influence on the radiosensitivity of vulva is used as the judgement of non-cell autonomy of DDR. Our previous study showed that pharynx is a radiosensitive somatic tissue [8,21], and there are some available pharynx-specific promoters such as P*myo-2* [28]. For these advantages, we choose pharynx as the DDR- rescued tissue. The pharynx of *C. elegans* is an isolated neuromuscular organ, and its primary function is to rhythmically pump, grind bacterial suspensions, and also sense environmental cues [29]. The promoter of the *myo-2* gene used here is primarily expressed in the muscular cells of the pharynx [28], thus, in the worm lines of pharynx-specific rescue, a functional DDR should be activated in muscle cells of the pharynx. Our previous study found that the posterior pharynx is radiosensitive to the induction of bystander effects [8]. Therefore, it is likely that in the worm lines of pharynx-specific rescue, the regulation of somatic DDR is initiated in muscular cells of the posterior pharynx. This gives two possibilities for the non-cell-autonomy of somatic DDR: one is that when the number of irradiated somatic cells, regardless of cell type, reaches a certain threshold value, they can synergistically contribute to a functional DDR; the other is that only certain key cells with residual DDR activity can implement thus functions. Therefore, it will be of importance to further examine the cell pattern of DDR activation in *C. elegans* in next work.

There is significant communication between the soma and germline in *C. elegans* [8,30,31]. Although the DDR of germline can be efficiently activated by irradiation exposure and other environmental toxicants, it is unclear why worms do not directly employ the DDR in the germline to distantly regulate DNA repair in the vulva. In this study, worms were exposed to ionizing irradiation at the stage of L1/L2, when each gonad contains only approximately 10 germ cells. Therefore, one possible reason is that the total DDR activity in the germline is not sufficient to implement a distant-range regulation. Another possibility might be that the genetic integrity of the reproductive system of *C. elegans* is mainly protected by DDR, whereas functions of somatic cells only support worms to physiologically survive [15], and thus they are protected through other mechanisms such as the innate immune system [30]. Even so, the evolutional retention of DNA repair capacity in somatic cells suggests that the non-cell-autonomous mode of somatic DDR might be initiated as a self-rescuing strategy, especially when DNA damage in somatic tissues is sufficient to deteriorate their physiological functions.

Bottom of DDR pathway are the apparatuses of cycle checkpoint, DNA repair, and apoptosis, all of which can modulate radiosensitivity of cells [1]. The radiation-induced death of vulval cells has been confirmed to be non-apoptotic [18]. In this study, it was shown that cycle checkpoints does also not participate in the responses of vulval cells to DNA damage (Figure 4A), and therefore, only DNA repair apparatus functions in this process. DDR kinases control DNA repair at three levels: DNA repair apparatus, chromatin environment, and cellular environment [32]. γ-rays can cause various types of DNA damage, including DNA double-strand breaks (DSB) that are typically repaired by the NHEJ and HR [33,34]. However, the γ-ray induced DNA damage in vulval cells of worms is mainly repaired by the NHEJ [19]. Moreover, the pharyngeal 9-1-1-ATR cascade regulates vulval radiosensitivity primarily through the NHEJ rather than the MMR and NER (Figure 4B and Figure A4). The selectivity for DNA repair patterns suggests that the pharyngeal DDR might distantly regulate DNA repair of vulva largely through the DNA repair apparatus. However, it is also possible that the control of DDR on DNA repair also occurs at the levels of chromatin and cellular environments, but the NHEJ might be more predominant in worm somatic cells.

Although a great progress has been made towards understanding the DDR, much remains to be investigated, especially in terms of how DDR proteins are activated and regulated. Here, we took advantages of the particularity of DDR in somatic cells of *C. elegans*, and put forward, and experimentally demonstrated the non-cell-autonomy of DDR. However, some limitations of *C. elegans* also affect the full dissection of the underlying mechanisms. For example, it is difficult to exclusively isolate the vulval cells from whole body of *C. elegans*, impeding the application of tissue-specific molecular analysis. For thus reasons, in addition to the NHEJ, no more downstream targets of DDR are well identified in this study. Therefore, they should be further explored using the cell-culture systems. Despite many differences in the developmental pattern and body structure from humans, *C. elegans* shares most of the genes and genetic mechanisms that govern development and disease in humans. In higher organisms, the differentiated cells and tissues, even tumors, usually have a heterogeneous activation of DDR due to the genetic and/or physiological inducements [3,35]. For example, many (possibly all) cancer cells lack one or more aspects of the DDR due to selective pressures operating during tumor evolution. Therefore, the non-cell autonomy of DDR might be of universality, which is required to be further inspected with various experimental systems.

## 4. Materials and Methods

### 4.1. Worm Strains and Growth

The *C. elegans* strain N2 variety Bristol was used for the general experiments. In addition, the following mutant strains were used: CB1487: *xpf-1(e1487)II*, IA123: *ced-25.1(ij48)I; unc-76 (e911) ijls10V*, JK1107: *glp-1(q224)III*, NL2550: *ppw-1(pk2505)I*, NW1613: *msh-2(ev679::Tc1)I*, RB2437: *cpr-4(ok3413)V*, RB647: *ced-25.3(ok358)III*, RB669: *wee-1.1(ok418)II*, RB864: *xpa-1(ok698)I*, RB873: *lig-4(ok716)III*, RB964: *cku-80(ok861)III*, TG2226: *xpc-1(tm3886)IV*, VC381: *atm-1(gk186)I*, WS2277: *hus-1(op241)I*, and WS2265: *hus-1(op244)I/hT2 [bli-4(e937) let-?(q782)qIs48](I;III)*. These strains were provided by the Caenorhabditis Genetics Center (St. Paul, MN, USA). The constructs of P*myo-2::gfp::hus-1::hus-1* 3’UTR and P*myo-2::gfp::atm-1::atm-1* 3’UTR were produced and inserted into the a pPD49_78 vector through the SphI and SpeI sites. The worm lines transgenic for extra-chromosome arrays P*myo-2::gfp::hus-1* and P*myo-2::gfp::atm-1* were produced through micro-injection of distal gonads by SunyBiotech (Fuzhou, China). Three independent integrant lines transgenic for P*myo-2::gfp::hus-1* were screened after 50 Gy of γ-irradiation, and then backcrossed twice with N2 worms to produce independent lines BPL1111, BPL1112, and BPL1113 [P*myo-2::gfp::hus-1; N2*], and the transgenic locus in line BPL1111 is on the X chromosome. These lines were further crossed with *hus-1* worms to produce the lines BPL1121, BPL1122, and BPL1123 [P*myo-2::gfp::hus-1;hus-1(op241)*], respectively. The independent integrant lines BPL1211 and BPL1212 transgenic for P*myo-2::gfp::atm-1;N2* and lines BPL1221 and BPL1222 [P*myo-2::gfp::atm-1; atm-1(gk186)*] were established using the same protocol. The transgenic locus in line BPL1121 is on the III chromosome. The negative control lines BPL1100 [P*myo-2::gfp; N2*], BPL1101 [P*myo-2::gfp; hus-1(op241)*], and BPL1201 [P*myo-2::gfp; atm-1(gk186)*] were also established using the same protocol. Worms were cultured and manipulated according to the standard procedures described by Brenner [36]. Briefly, worms were cultured at 20 °C on nematode growth media (NGM) [3 g/L NaCl (cat no: A501218, Sangon Biotech, Shanghai, China), 17 g/L AgarB (cat no: A610012, Sangon Biotech), 2.5 g/L peptone (cat no: A505247, Sangon Biotech), 2.71 g/L $KH_{2}PO_{4}$ (cat no: A501211, Sangon Biotech), 0.89 g/L $K_{2}HPO_{4}$ (cat no: A501212, Sangon Biotech), 0.11 g/L $CaCl_{2}$ (cat no: A501330, Sangon Biotech), 0.12 g/L $MgSO_{4}$ (cat no: A601988, Sangon Biotech) and 5 mg/L cholesterol (cat no: A100433, Sangon Biotech)], supplemented with *E. coli* OP50 as a food source. The *glp-1* worms were cultured at 15 °C for normal growth and 25 °C for induction of the germless phenotype [37].

### 4.2. Synchronization of Worms and γ-Irradiation Treatment

Synchronized larvae were obtained according to previously described methods [18]. Briefly, gravid hermaphrodites were washed off plates with M9 solution [5 g/L NaCl (cat no: A501218, Sangon Biotech), 6 g/L $Na_{2}HPO_{4}$ (cat no: A501727, Sangon Biotech), 3 g/L $KH_{2}PO_{4}$ (cat no: A501211, Sangon Biotech), 0.12 g/L $MgSO_{4}$ (cat no: A601988, Sangon Biotech)] and lysed with 200 g/L NaOH (cat no: 10019718, SINOPHARM, Beijing, China) and bleach solution (active
chlorine≥5.2%, cat no: 80010428, SINOPHARM). The resulting embryos were washed with M9, plated on unseeded agarose-containing petri dishes, and left at 20 °C for 14 h. Larvae were transferred to OP50-seeded plates, or 24-well culture plates for liquid culture of conditioned medium. The placement of food was considered to be 1 h old. Irradiation of worms was carried out using a Biobeam $Cs^{137}$ irradiator (cat no: GM2000, Gamma-Service Medical, Leipzig, Germany) at a dose rate of 3.37 Gy/min. After irradiation, the worms were transferred onto fresh OP50-seeded plates. The same worms placed next to the irradiator were used as control of sham-irradiation. N2 worms were irradiated at 16 h. In order to balance the vulval radiosensitivity, the irradiation timing of other strains was individually determined when their body length was identical to that of 16 h N2 worms. The assayed worms were usually grown on petri dishes with a diameter of 60 mm, and culture density is approximately 250 worms per dish.

### 4.3. Phenotypic Characterization of Worm Vulva

Phenotypic characterization of vulva was performed as previously reported [18], with minor modifications. Worms were anesthetized with 40 mM $NaN_{3}$ (cat no: S2002, Sigma, St. Louis, MO, USA), placed onto glass slides with some M9 solution and examined using a 20× or 40× optic microscope (SMZ168, Motic, Xiamen, Fujian, China). The ratio of worms with protruding vulva was scored at adult stage, approximately 3 days after irradiation. The final data represent the average of at least 3 independent experiments, and at least 200 worms were used for each experiment. The proportions of protruding vulva of all strains were checked after 100 Gy of sham-irradiation, and no obvious change was observed, as shown in Table A1.

### 4.4. Co-Culture Experiments of Worms

The co-culture experiments were carried out on OP50-seeded petri dishes (Φ = 60 mm), where two kinds of worms were cultured together at a ratio of 1:1. However, the ratio was also adjusted for specific purposes. They were distinguished from each other according to whether there is GFP fluorescence in the pharynx. The proportions of protruding vulva of worms co-cultured with the related strains were checked after 100 Gy of sham-irradiation, and they are not significantly different from the worms cultured alone (marked as Blank), as shown in Table A2.

### 4.5. Production of Conditioned Medium and Treatment with Enzymes and Inhibitors

The conditioned medium was prepared according to a previously described protocol [9]. A total of 1.5 mL of OP50 was added into 30 mL of S-medium (5.85 g/L $KH_{2}PO_{4}$ (cat no: A501211, Sangon Biotech), 5.70 g/L NaCl (cat no: A501218, Sangon Biotech), 0.97 g/L $K_{2}HPO_{4}$ (cat no: A501212, Sangon Biotech), 4.87 mg/L cholesterol (cat no: A100433, Sangon Biotech), 0.19 g/L Citric acid monohydrate (cat no: A502123, Sangon Biotech), 2.86 g/L Potassium citrate monohydrate (cat no: A500767, Sangon Biotech), 20.08 mg/L Ethylenediaminetetraacetic acid disodium salt (cat no: A100105, Sangon Biotech), 6.73 mg/L $FeSO_{4}$${}^{.}7H_{2}O$ (cat no: A600461, Sangon Biotech), 1.95 mg/L $MnCl_{2}$${}^{.}4H_{2}O$ (cat no: A500331, Sangon Biotech), 2.83 mg/L $ZnSO_{4}$${}^{.}7H_{2}O$ (cat no: A602906, Sangon Biotech), 0.24 mg/L $CuSO4^{.}5H_{2}O$ (cat no: A600063, Sangon Biotech), 0.32 g/L $CaCl_{2}$ (cat no: A501330, Sangon Biotech) and 0.35 g/L $MgSO_{4}$ (cat no: A601988, Sangon Biotech) in 50 mL Corning centrifuge tubes. Each tube contained approximately 7000 worms. The worms were grown on a shaker at 20 °C with a speed of 200 rpm. After 15 h of culture, the worms were irradiated with γ-rays. After another 24 h of culture, these tubes were centrifuged at 5000× *g* for 15 min and the conditioned medium was obtained through filtration of the medium with a 0.22-μm filter. 1ml of conditioned medium was mixed with 10 μL DNase (1 UμL${}^{-1}$) (cat no: B300066, Sangon Biotech), 100 μL RNase (10 μg μL${}^{-1}$) (cat no:B300068, Sangon Biotech) or 10 μL trypsin (5 μg μL${}^{-1}$) (cat no: T2610, Sigma) solutions for 1 h at 30 °C, respectively. The treated or untreated conditioned medium was then used to culture the irradiated animals at 20 °C in a 24-well culture plate until the vulva is fully developed. The CPR-4 inhibitors baicalein (cat no: 465119, Sigma) and CA-074 (cat no: HY-103350, MedChemExpress, Shanghai, China) were dissolved in dimethyl sulfoxide (DMSO) (cat no: A610163, Sangon Biotech) and then added into the conditioned medium. The final concentration of DMSO in the conditioned medium did not exceed 0.1%. The irradiated worms were transferred into conditioned medium containing 10 μM CA-074 or 250 μM baicalein until detection of vulva malformation [25].

### 4.6. RNAi Experiments of Worms

RNAi experiments were performed using a bacterial feeding protocol [38]. HT115 bacteria expressing the empty vector L4440 and bacterial clones expressing dsRNA (Ahringer RNAi library) were presented by the laboratory of Guang Shouhong at the University of Science and Technology of China [39]. The specificity of RNAi was validated by phenotypic comparison between RNAi knockdowns and their corresponding mutants [40]. Next, approximately 2000 L4 worms were grown on NGM plates with 0.5 mM isopropyl β-d-1-thiogalactopyranoside (IPTG) (cat no: A100487, Sangon Biotech, Shanghai, China) and specified RNAi bacteria until their eggs could be synchronized. The synchronized progeny worms were transferred onto NGM plates with IPTG and dsRNA-expressing bacteria for subsequent irradiation treatment and detection of vulval malformation.

### 4.7. Detection of Worm Growth and Development

16-h old worms were first irradiated with γ-rays, and then grew until adult stage. Subsequently, 150 adult worms were photographed, and body length was measured using the Image J program (1.8.0 version, NIH, Bethesda, MD, USA). For the detection of body development, the worms were first stained with DAPI (cat no: E607303-0002, Sangon Biotech, Shanghai, China) according to a previously described protocol [27]. The gonads of worms were checked using a fluorescence microscope (DM4B, Leica, Wetzlar, Germany), and the worm development was evaluated according to the ratio of worms with U-shaped gonads to those with linear gonads. The final data represent the average of at least three independent experiments, and at least 200 worms were used for each experiment.

### 4.8. Statistical Analysis

All results are presented as the mean ± standard deviation. The significance of the experiments was confirmed by Student’s *t*-test using origin program (OriginPro 2018C(64-bit) SR1 b9.5.1.195(education), OriginLab, Northampton, MA, USA) and by nonparametric test (SPSS, 23.0 version, IBM, New York, USA) only for the data of body length with non-normal distribution. Significance was considered at *p* < 0.05, marked as * *p* < 0.05 and ** *p* < 0.01.

## 5. Conclusions

In this study, we presented the experimental evidence that the DDR can operate in a non-cell autonomous mode, and the CPR-4 mediates the non-cell autonomy of DDR. The non-cell autonomy of DDR is relevant to the tumor micro-environment, and its discovery might provide new cues for cancer prevention and treatment.

## Figures and Tables

**Figure 1 ijms-23-07544-f001:**
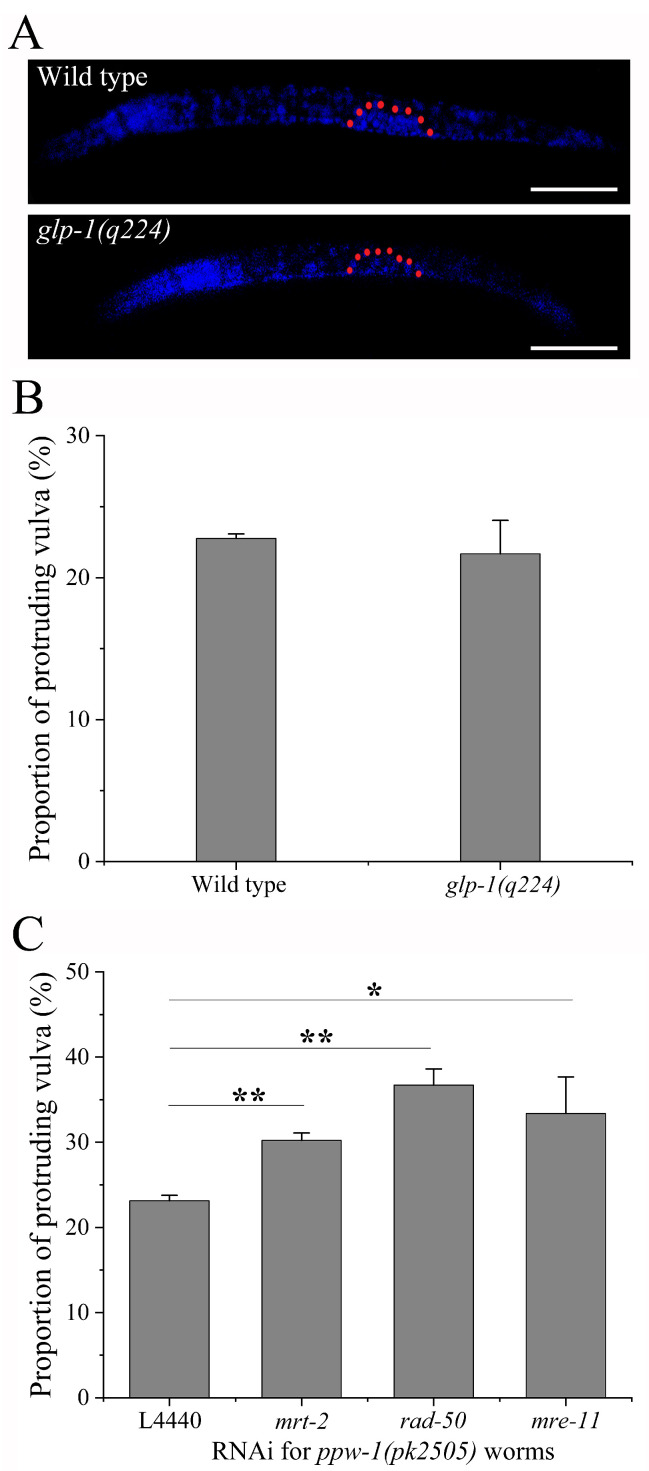
Participation of somatic DDR in regulating vulval radiosensitivity of *C. elegans*: (**A**) DAPI staining of N2 and *glp-1* worms at the age of 13 h at 25 °C, and gonad region is outlined by the dotted red line, scale bar: 50 μm; (**B**) the radiosensitivity of reproductive death of vulval cells in *glp-1* worms, which is evaluated by the proportion of protruding vulva; (**C**) the vulval radiosensitivity of *ppw-1* worms subjected to RNAi of the *mrt-2*, *mre-11*, and *rad-50* genes, respectively. Results are means ± SD (*n* ≥ 3, *t*-test, * *p* < 0.05 and ** *p* < 0.01).

**Figure 2 ijms-23-07544-f002:**
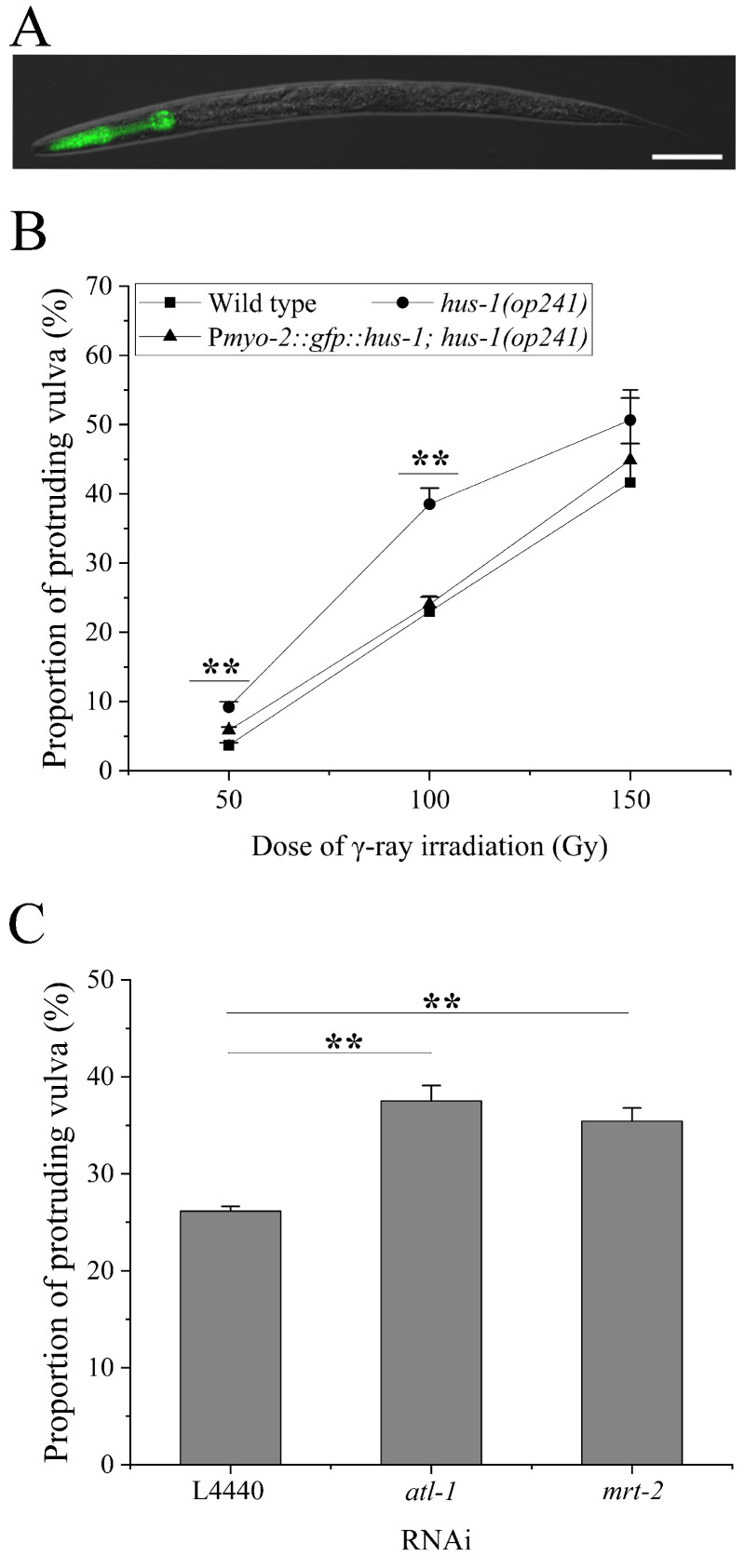
Inter-tissue regulation of pharyngeal ATR cascade on vulval radiosensitivity: (**A**) Pharynx-specific expression of the *hus-1* gene in worm line BPL1121: P*myo-2::gfp::hus-1; hus-1(op241)*, scale bar: 50 μm; (**B**) the vulval radiosensitivity of BPL1121 worms after exposure to the indicated doses of γ-irradiation; (**C**) the vulval radiosensitivity of BPL1121 worms subjected to RNAi of *atl-1* and *mrt-2* genes Results are means ± SD (*n* ≥ 3, *t*-test, ** *p* < 0.01).

**Figure 3 ijms-23-07544-f003:**
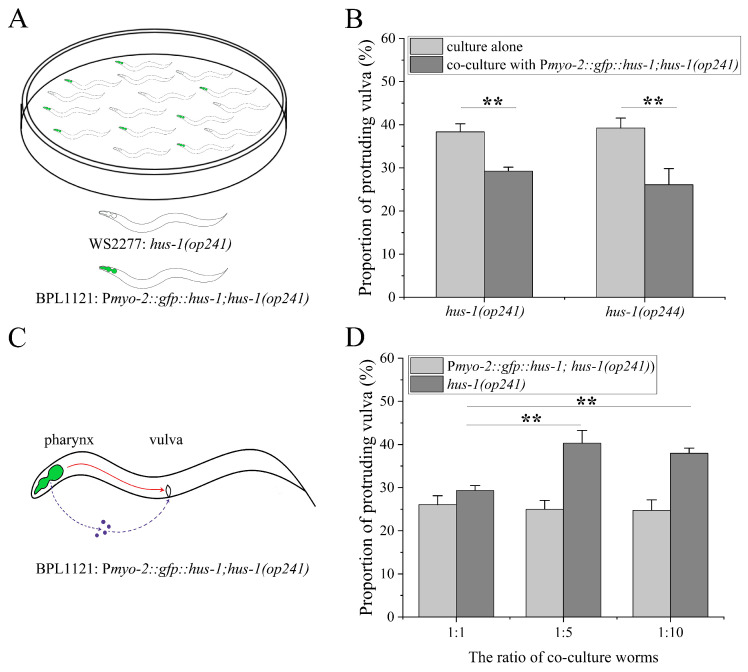
Inter-individual regulation of pharyngeal ATR cascade on vulval radiosensitivity: (**A**) the schematic diagram of co-culture experimental system of worms, in which BPL1121: P*myo-2::gfp::hus-1; hus-1(op241)* and *hus-1* worms are distinguished according to GFP fluorescence in the pharynx; (**B**) the vulval radiosensitivity of *hus-1(op241)* and *hus-1(op244)* worms co-cultured with BPL1121 worms; (**C**) two possible transmitting/cascading modes of pharynx-derived damage signals in BPL1121 worms, represented schematically by the red line and dotted blue lines, respectively; (**D**) the vulval radiosensitivity of *hus-1* worms co-cultured with BPL1121 worms at the indicated ratios. Results are means ± SD (*n* ≥ 3, *t*-test, ** *p* < 0.01).

**Figure 4 ijms-23-07544-f004:**
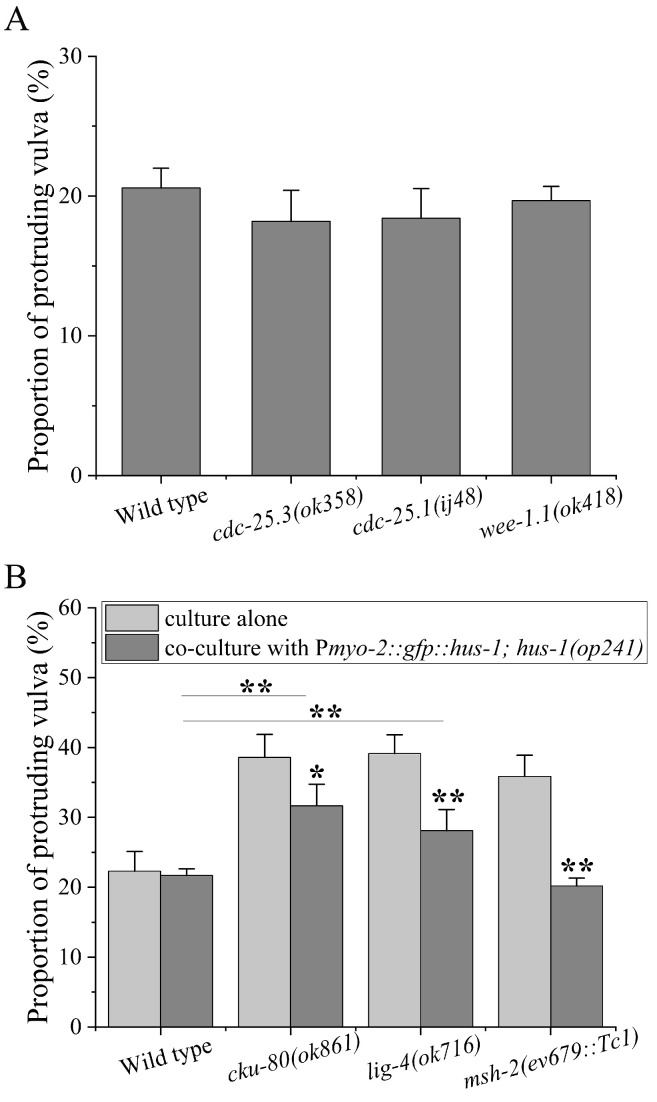
Regulation of pharyngeal ATR cascade on NHEJ repair of the vulva: (**A**) the vulva radiosensitivity of *cdc-25.1*, *cdc-25.3*, and *wee-1.1* worms; (**B**) the vulval radiosensitivity of *cku-80*, *lig-4*, and *msh-2* worms co-cultured with worm line BPL1121: P*myo-2::gfp::hus-1; hus-1(op241)*. Results are means ± SD (*n* ≥ 3, *t*-test, * *p* < 0.05 and ** *p* < 0.01).

**Figure 5 ijms-23-07544-f005:**
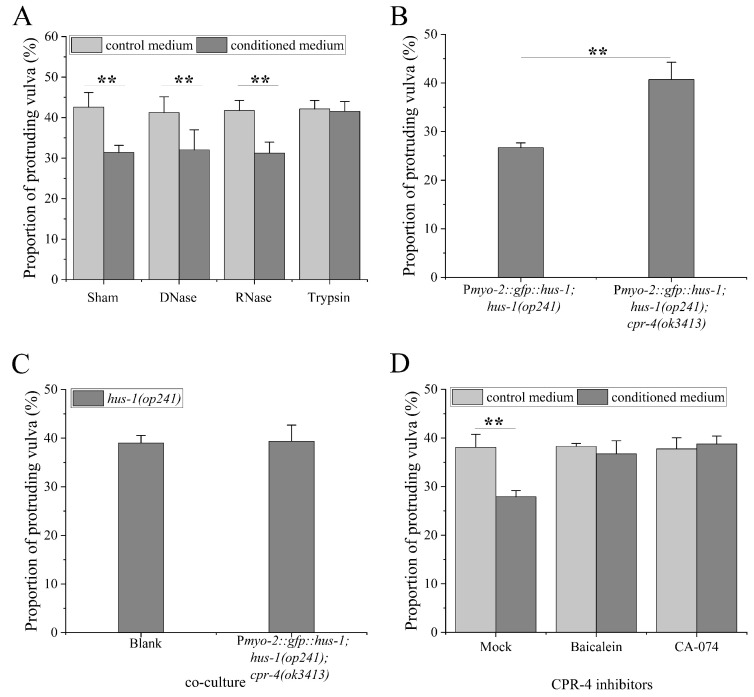
Role of CPR-4 in inter-tissue regulation of the pharyngeal ATR cascade: (**A**) the vulval radiosensitivity of *hus-1* worms cultured in conditioned medium treated with DNase, RNase, and trypsin; (**B**) the vulval radiosensitivity of worm line BPL1131: P*myo-2::gfp::hus-1; hus-1(op241); cpr-4(ok3413)*; (**C**) the vulval radiosensitivity of *hus-1* worms co-cultured with BPL1131 worms, Blank: culture alone; (**D**) the vulval radiosensitivity of *hus-1* worms cultured in the conditioned medium treated with CPR-4 inhibitors baicalein or CA-074. Results are means ± SD (*n* ≥ 3, *t*-test, ** *p* < 0.01).

**Figure 6 ijms-23-07544-f006:**
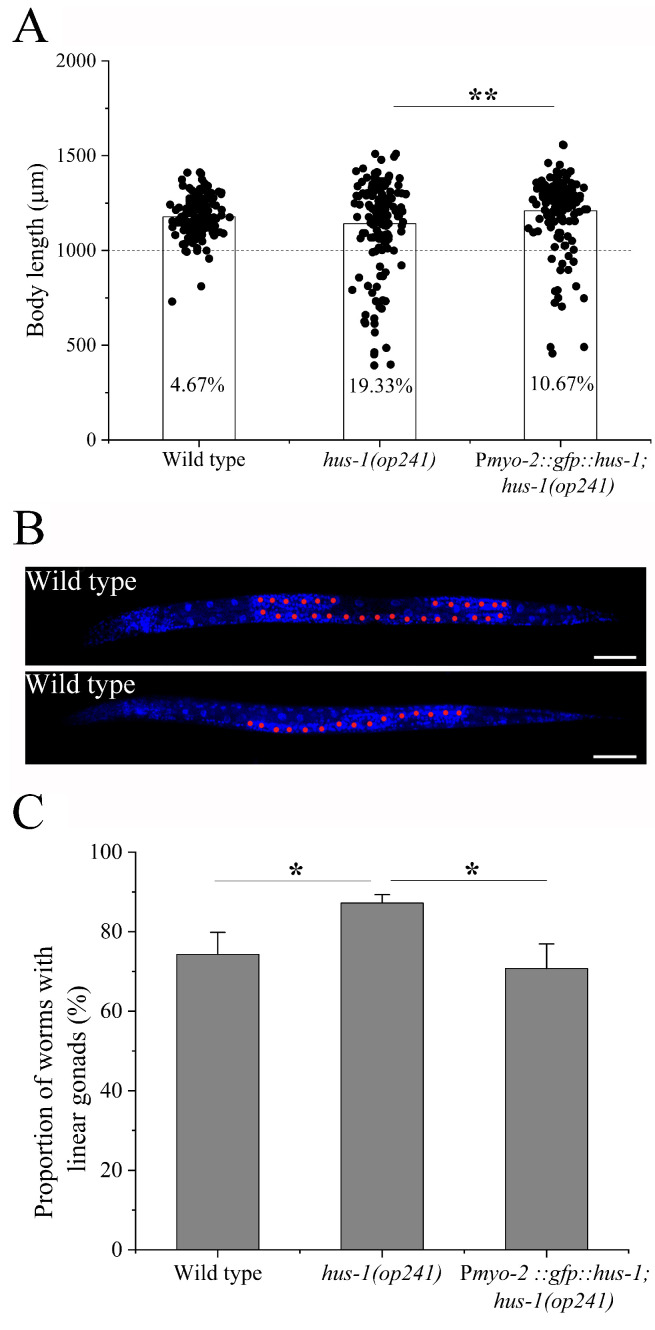
Systemic regulation of pharyngeal ATR cascade on worm growth and development: (**A**) body length of adult worms BPL1121: P*myo-2::gfp::hus-1; hus-1(op241)* after γ-irradiation (*n* = 150), and the numbers in datum bars are the percentage of worms with body length of less than 1 mm.; (**B**) N2 worms with U-shaped (top) and linear (bottom) gonads, scale bar: 50 μm; (**C**) the proportion of worms with linear gonads at 32 h after γ-irradiation. Results are means ± SD (*n* ≥ 3, *t*-test, nonparametric test for panel A, * *p* < 0.05, ** *p* < 0.01).

**Figure 7 ijms-23-07544-f007:**
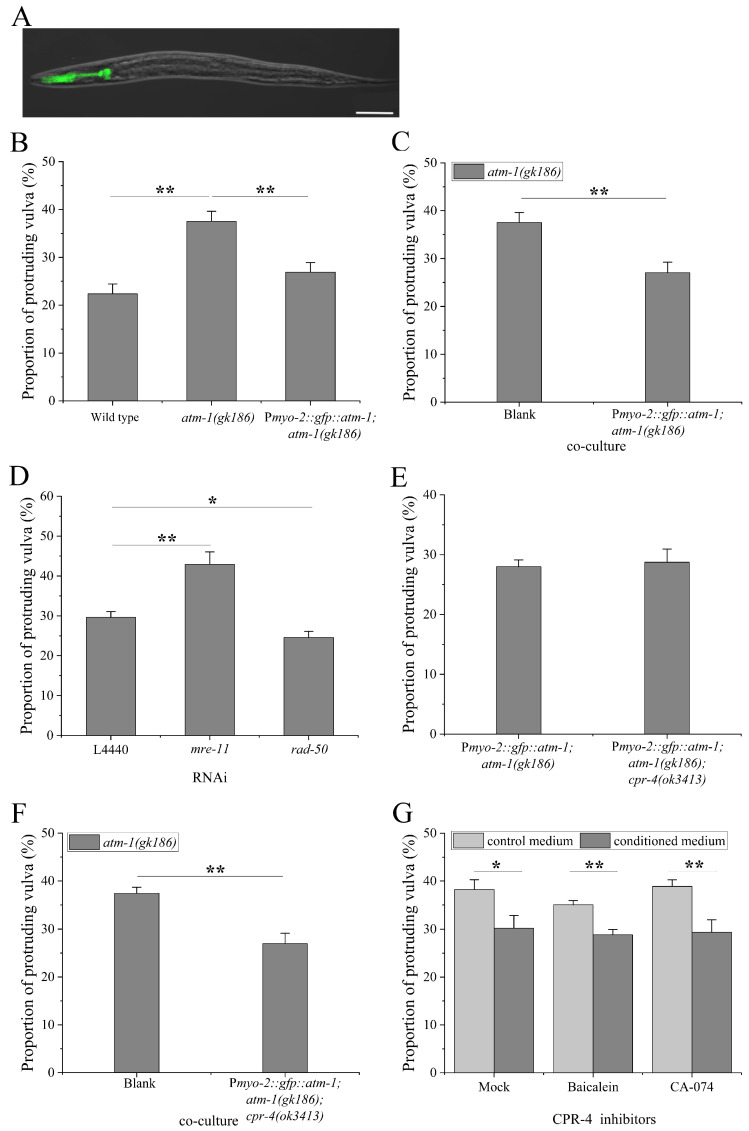
Inter-tissue regulation of pharyngeal ATM cascade on vulval radiosensitivity: (**A**) pharynx-specific expression of the *atm-1* gene in worm line BPL1221: P*myo-2::gfp::atm-1; atm-1(gk186)*, scale bar: 50 μm; (**B**) the vulval radiosensitivity of BPL1221 worms; (**C**) the vulval radiosensitivity of *atm-1* worms co-cultured with BPL1221 worms, Blank: culture alone; (**D**) the vulval radiosensitivity of BPL1221 worms subjected to RNAi of *mre-11* and *rad-50* genes; (**E**) the vulval radiosensitivity of worm line BPL1231: P*myo-2::gfp::atm-1; atm-1(gk186); cpr-4(ok3413)*; (**F**) the vulval radiosensitivity of *atm-1* worms co-cultured with BPL1231 worms, Blank: culture alone; (**G**) the vulval radiosensitivity of *atm-1* worms cultured in the conditioned medium of BPL1221 worms treated with CPR-4 inhibitors baicalein and CA-074, respectively. Results are means ± SD (*n* ≥ 3, *t*-test, * *p* < 0.05 and ** *p* < 0.01).

## Data Availability

Data and materials will be made available upon reasonable request.

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
