# Peer review of "A Non-Cell-Autonomous Mode of DNA Damage Response in Soma of Caenorhabditis elegans"

_ijms, 2022, doi:10.3390/ijms23147544_

Round 1

Reviewer 1 Report

However the paper is interesting and in general well-designed and written, some changes are mandatory.

1. The introduction has to be ended by "the aim of the study". Please add.

2. In the statistical analysis please add the name of the software (also company, country); had you made the Shapiro-Wilk's test before the t-Student test's was performed?

3. the results section is a mix of the results and discussion section and vice versa. It should be corrected. 

4. Please add the strengths and limitations of this study in the discussion section.

5. Author Contributions statement should be corrected according to the Journal's recommendation.

6. References have to updated; adding more papers from 2017.

7. Some typo mistakes need to be corrected.

8. in the abstract please add more information about obtained data.

Reviewer 2 Report

The manuscript “A Non-Cell-Autonomous Mode of DNA Damage Response in Soma of Caenorhabditis elegans” by Dai et al. presents interesting data on non-cell-autonomy role of DDR pathways/repair factors. Since growing evidence suggests that some DNA damage repair factors could affect and regulate the response to DNA damage in other cells or physically distant tissues, this could be relevant to the tumor micro-environment.

The authors using various C. elegans lines with tissue-specific rescue of DDR demonstrated that the 9-1-1-ATR and MRN-ATM cascading pathways in pharynx distantly regulate the DNA damage repair in vulva. They found CPR-4, a cysteine protease cathepsin B, as an important factor that has a pivotal role in mediating the inter-tissue and inter-individual regulation of DDR.

Overall, the manuscript is well written. The experiments were designed and conducted properly. The results are presented clearly. From my view the manuscript is complete and requires no additional experiments or controls.

Reviewer 3 Report

The manuscript describes C. elegans model system used for the radiosensitivity experiments. It points out the possibility that the DNA repair response can be regulated in a non-cell-autonomous mode, and the distant organs can influence each other's response to the gamma-irradiation, which is known to induce DNA damage.

There are several major points to consider:

1) Materials and methods section needs to be expanded. The methods will benefit from a more precise and detailed description. Many materials lack producer information and/or catalog numbers. Altogether it makes the experiments harder to interpret and reproduce.

2) Figures are made using a small and hardly readable font. The most the Figures will benefit from using larger fonts.

3) The manuscript is written in an understandable way, although there are typos (e.g. missing spaces, parentheses), repeats, and overall impression that it is not written using a research article style. It lacks clarity and sharpness. I was not convinced that the results support the conclusions at this point.

Round 2

Reviewer 1 Report

the paper has been improved enough.

Author Response

Thanks a lot. 

Reviewer 3 Report

The original manuscript is revised and the presentation is improved. Not all of the original questions have been followed up, e.g., materials and methods still missing clarification of origin of reagents, or concentrations (for example, lines 317-318). 

How the digestion is performed by NaOH? Should it be called lysis?

It is not clear how the specific effect of RNAi was evaluated, and how the gene expression was controlled.

The authors emphasize that the discovery is very new and unique, suggesting that it is potentially of very high impact, although it needs proper controls at all stages to ensure that the message is accepted by the research community.

Round 3

Reviewer 3 Report

The authors revised the manuscript based on the reviewer's suggestions and answered the key questions.